# Design and Research of Automatic Garment-Pattern-Generation System Based on Parameterized Design

**Peng Jin [1], Jintu Fan [2,3,*], Rong Zheng [2,*], Qing Chen [2], Le Liu [4], Runtian Jiang [1] and Hui Zhang [1]**

1   College of Fashion and Design, Donghua University, Changning District, Shanghai 200051, China
2   Shanghai International Fashion Innovation Centre, Donghua University, Changning District,
    Shanghai 200051, China
3   Institute of Textiles and Clothing, The Hong Kong Polytechnic University, Kowloon, Hong Kong, China
4   School of Design, Jiangnan University, Wuxi 214122, China
*   Correspondence: jin-tu.fan@polyu.edu.hk (J.F.); rzheng@dhu.edu.cn (R.Z.)

**Abstract:** Personalization in the apparel industry shows importance and the potential for demand, but the existing personalization has unreasonable time cost, labor cost, and resource waste. To solve the problems of the waste of resources as well as both time and labor cost caused by manual pattern making in clothing personalization, a method of automatic garment pattern generation based on a parametric formula and the Python language was proposed. Based on the classification of common curves in patterns, three curve fitting algorithms based on different parameters were derived and combined with the Python language to achieve personalized generation of different patterns by classifying the parameters in the system into key parameters, secondary parameters, and variable parameters. Three different methods for verifying the accuracy of the garment patterns were proposed based on curve fitting similarity and three-dimensional virtual modeling, and the accuracy of the proposed system was verified. The results show that the accuracy and comfort of the patterns generated via the system were high. Meanwhile, the Python-language-based system fits well with the production system of enterprises, which can improve the rapid response capability of garment personalization, greatly save the time cost and labor cost of enterprises, reduce resource loss, and contribute to the sustainable development of the garment industry.

**Keywords:** garment pattern; automation; parametric design; Python; personalization

## 1. Introduction

The traditional garment production mode includes mass production mode and personalized production mode [1]. With the continuous improvement of material and cultural living standards, the traditional mass production mode has been increasingly unable to meet the fashionable, diversified, and personalized needs of the consumer [2]. Although the customized clothing production mode can meet the above requirements, the time cost and labor cost are high, and there is a waste of resources, which is contrary to the concept of sustainable fashion development. In the process of garment customization, the measurement, cutting, sewing, and storage are all highly automated [3–5], but the pattern-making process is still dominated by traditional manual pattern making, such as the use of Fuyi, Bock, and other computer aided design (CAD) software [6–8].

Although garment CAD technology has been widely used in garment structure design, it still relies heavily on personal experience in actual production, and designers who lack experience in pattern making cannot directly participate in pattern making. According to the two-dimensional paper pattern, designers often cannot intuitively feel the three-dimensional garment form; the traditional method is to make three-dimensional sample garments through the blank and then adjust the paper pattern, but this process usually needs to be repeated several times, which is very tedious. Stereo cutting is not easy to control the direction of fabric lines, and the operation is tedious, with high time and labor

costs; in addition, mass production is only suitable for standard body types and cannot meet the needs of individualized garments, while tailoring meets the needs of individualized garments, however, the workload is large and inefficient. Therefore, the realization of an automated garment pattern-making system plays an important role in the efficient, green, and sustainable development of the garment industry.

With the continuous development of computer technology, expert systems [9,10], machine learning [11,12], and other artificial intelligence techniques [13–15] have laid a good technical foundation for garment pattern making. It has been envisioned that the future of computerized pattern making is based on computer recognition and an understanding of garment styles, with the computer reasoning and judging the pattern making and the operator only responsible for judging and correcting. Based on this, in recent years, a series of research have been conducted for the intelligent production of garment patterns. Papachristou et al. [16] analyzed and compared traditional manual pattern making and existing CAD pattern design systems, as well as proposed user-friendly features and function collection modules for CAD pattern design systems. Ho et al. [17] introduced the modeling method of artificial neural networks, described the current state of application of the technique in comfort research, and discussed the technical structure of the comfort neural network. Celcar et al. [18] discussed the key techniques for developing an expert system for automatic pattern generation in three aspects: the description of garment structure, acquisition of expert knowledge and knowledge base, and adoption of parametric design ideas. Wang et al. [19] proposed a knowledge-supported garment pattern design method based on fuzzy logic and artificial neural networks by learning the knowledge of experienced pattern makers. Ma et al. [20] proposed an intelligent garment paper pattern design method based on BP neural networks but did not conduct experiments to verify the accuracy of the method.

In addition to the above methods, some researchers have also proposed methods for generating garment patterns from the perspective of parametric design [21,22]. At present, there are mainly two ways of utilizing parametric pattern making for garments: the first one is to use the upgraded garment-specific CAD mapping software or three-dimensional modeling software; the other one is to use the secondary developed function to drive other general drawing or digital image processing software, such as mapping with the MatLab data visualization function, AutoCAD parametric drawing, etc. For example, by determining the basic dimensions of the human body, Sybille K proposes a parametric modeling method built into the three-dimensional design environment that automatically changes the dimensions of the model and further derives it into a two-dimensional prototype [23]. Gu et al. proposed an automatic curve drawing method based on Bessel curve control points by studying the process of garment structure curves but did not practice on specific parametric applications [24]. By using the MATLAB programming software to generate artistic graphics that fit a specific style, Wang et al. combined Photoshop with the image processing software to process and redesign the generated graphics and apply them to the garment pattern [25].

Based on the above analysis, it can be seen that various programming techniques or software used to realize the automatic pattern making of garments has largely changed the requirements of human resources for traditional garment pattern making, and to a certain extent has improved the production efficiency of enterprises and provided new ideas for the research of garment-intelligent manufacturing.

However, there are still some difficulties to be overcome, such as: (1) limited by the problem of mapping logic, the current approach targets a single digital image processing software (e.g., Matlab or AutoCAD), which is poorly adapted to different drawing software and difficult to adapt to the production requirements of different companies; (2) the existing research lacks verification experiments, and the accuracy of curve fitting and automatic pattern generation methods is difficult to guarantee; (3) machine learning-based garment pattern making system is highly dependent on the existing pattern library, which is difficult to extend to other types or styles of garments, and has poor adaptability to people of

different body types; (4) the code is long and difficult to match with the original software and hardware systems of enterprises; (5) the input of data variables cannot interact with external ports, and some parameter data cannot be stored, and the overall integration of the system is low.

Accordingly, based on the design concept of parametric design, a method of automatic garment pattern generation based on parametric variables and the Python language was proposed in this paper, in order to solve the problem of wasted resources and costs caused by manual pattern making in garment personalization.

## 2. Method

In this paper, an automatic garment pattern generation method based on the parametric design concept was proposed based on parametric variables and the Python language. Parametric design is a constraint-based product modeling method, which is a set of sequences that constrain and describe the structure and dimensions of geometry through the use of parameters [26]. By changing the constraints and giving different values for the parameter sequence, the constraints can be driven to obtain new target geometries and quickly generate design solutions for different products. In apparel pattern design, factors such as size and structure are crucial constraints that determine the final product design. The internal structure of the garment paper pattern is interrelated, and changes in the parameters of the dimensions cause corresponding changes in the parameters of the related components. The consistency and bidirectionality of local and overall changes allow for the rapid regeneration of the design solution without the need for manual adjustments and modifications to all details.

Python is a simple, easy-to-learn, and powerful programming language widely used in web development, artificial intelligence development, scientific computing, software development, and data processing analysis. Because the Python language can import third-party libraries, there is no need to write additional underlying logic for formulas or algorithms, so complex model operations can be performed using lightweight code [27].

The overall research methodology of this paper is as follows: the garment pattern was decomposed, and the decomposed pattern was represented in a two-dimensional plane using parametric variables. Based on the classification of common curves in the garment pattern, three curve fitting algorithms based on different parameters were derived and combined with the Python language to design curves that can be recognized using the CNC (Computerised Numerical Control Machine) cutting equipment in the workshop. Then, the size and constraint relationship between the structure line of the garment pattern and the key parts of the human body were analyzed to establish the parametric coordinate system of the pattern. After the parametric coordinate system was completed, the algorithm was described for the pattern and a complete parametric algorithm model for women's shirts was constructed, which contained a curve function layer, a pattern function layer, and a function call layer. If the parameters need to be adjusted, it is only necessary to change the size of the variables in the model, and automatic generation of the pattern can be realized by using the parameter-driven model without large-scale modification of the basic pattern. A parametric pattern generation system was developed for a women's shirt that matches the design style of a certain company, and the accuracy of the system-generated pattern was verified by comparing the similarity between the manual pattern and the system-generated pattern of several subjects with different body types, and three-dimensional software was used to conduct virtual tests to verify the comfort of the patterns. By modifying the parametric variables in the system, the system can generate and call for new models based on the original parametric models to meet the demands of garment enterprises for the automated generation of personalized patterns for different body types of people and different types (or styles) of garments; improve the production efficiency of enterprises; reduce the waste of resources; and meet the sustainable development of the garment industry. As shown in Figure 1.

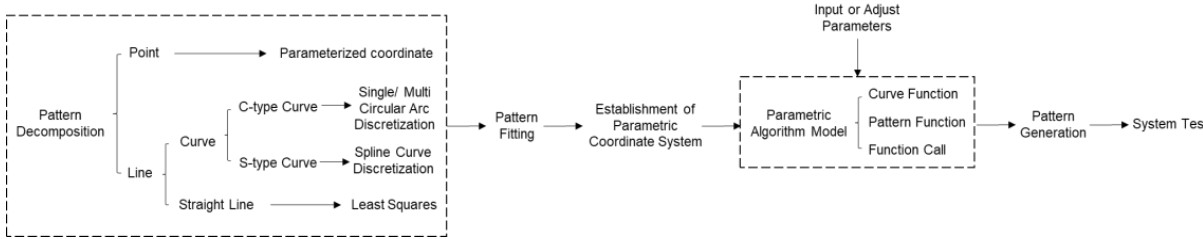

**Figure 1.** Research Method.

*2.1. Pattern Decomposition*

A garment pattern can be divided into geometric figures composed of points, lines, and curves. In the process of pattern drawing, it is usually necessary to determine the functional relationship between points and lines with the help of a right-angle coordinate system. In this paper, the horizontal axis of the coordinate system is the *x*-axis, and the vertical axis is the *y*-axis.

### 2.1.1. Points

In the plane coordinate system, the coordinate position of a point (a, b) can be expressed as $(x - a)^2 + (y - b)^2 = 0$. Since a, b are parametric variables, when changing the style of a pattern, it is only necessary to change the coordinates of certain points. This parametric variables-based conversion mode can greatly improve the efficiency of converting garment styles, reduce arithmetic power loss, and improve the efficiency of pattern making.

### 2.1.2. Line

The lines in the garment pattern can be divided into straight lines and curves, of which the curves can be further divided into C-type curves and S-type curves. C-type curves can be divided into a single intersection basic-type curve or multiple single intersection basic-type curve; S-type curves are mainly composed of two reverse single intersection basic-type curves. Some of the C-type curves and S-type curves in the garment pattern are shown in Table 1.

**Table 1.** Classification of lines in garment patterns.

| | Category | | Illustration | Fitting Method |
|---|---|---|---|---|
| | **Straight Line** | | \ | **Least Squares** |
| Curve | C-type | Single intersection | | Single-circular-arc discretization |
| | | Multiple intersection | | Multiple-circular-arc discretization |
| | S-type | | | Spline curve discretization |

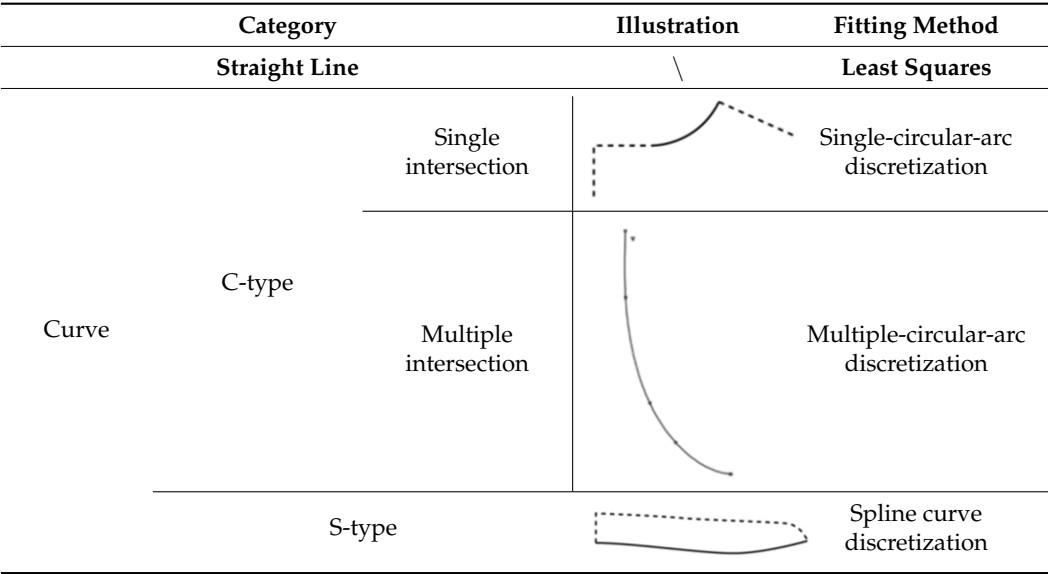

*2.2. Pattern Fitting*

2.2.1. Straight-Line Segment Fitting

In the plane coordinate system, a unique straight-line segment can be determined between two points, and the least squares method was used to fit the straight-line segment in this paper.

2.2.2. Curve Fitting

Curve fitting methods are more complex than straight line fitting. Based on the AutoCAD drawing software and the pyautocad library, the arc and spline curve tools were used to fit the contour curve of the garment pattern. A polyline matrix was created to discretize the curves into micro-line segments, which can be recognized using the CNC cutting equipment as pattern curves.

(1) Single-circular-arc discretization

Single-circular-arc discretization is applicable to the drawing of single intersection basic-type curve in C-type curves, such as posterior collar nest arcs, rounded collar curves, etc. By introducing a third parameter (bulge) between two parameterized coordinate points, a unique single-circular-arc curve passing between those two coordinates can be determined. The bulge reflects the size of the corresponding arc between the two points and the direction of the arc; according to geometric knowledge, the value of the bulge is the tangent of 1/4 of the angle contained in this arc, and the bulge size can be derived from the bulge formula, as shown in Equation (1). At the same time, the arc discrete function is constructed and the arc angle $\theta$ and radius $r$ are calculated, the arc np matrix is established, the starting point variable and the ending point variable of the arc are input, the start point variable $P_{start}(x_{start}, y_{start})$ and the end point variable $P_{end}(x_{end}, y_{end})$ of the arc are input, and the arc angle and radius are derived from the arc bulge. As shown in Figure 2a and Equations (1) and (2):

$$\theta = arctan(|bluge|) \times 4 \tag{1}$$

$$d = \sqrt{(start_x - end_x)^2 + (start_y - end_y)^2} \tag{2}$$

where, $\alpha$ is the arc circle center angle; $\theta$ is the number of radians of the chord length, $d$ is the distance between $P_1$ and $P_2$; $start_x$ and $end_x$ are the $x$-axis values of $P_1$ and $P_2$; $start_y$ and $end_y$ are the $y$-axis values of $P_1$ and $P_1$.

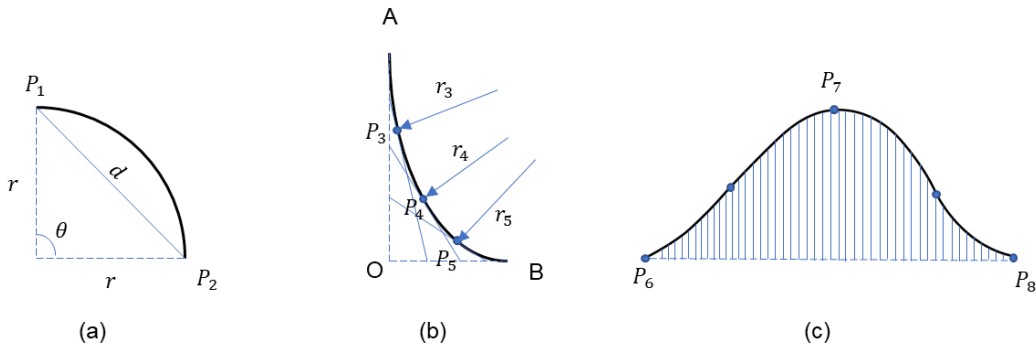

(a)       (b)       (c)

**Figure 2.** Schematic diagram of each type of curve fitting algorithm. (**a**) Single-circular-arc discretization; (**b**) multiple-circular-arc discretization; (**c**) Spline curve discretization.

According to the triangle rule of vector addition, the positive and negative values of the $x$-axis and the direction of the arc can be determined. The arc is decomposed into $n_1$ segments ($n_1$ is an integer and $n_1$ is greater than zero; the larger the n1 number, the better the fit), and the coordinates of the endpoints of each segment are derived using the trigonometric formula and the values are inserted into the matrix to output the

multi-segment arc matrix after bulge discretization. The coordinates equation is shown in Equations (3)–(6).

When bulge > 0.

$$y = r \times \sin(\theta_0 - i \times \theta_{step}) \tag{3}$$

$$x = r \times \cos(\theta_0 - i \times \theta_{step}) \tag{4}$$

When bulge < 0.

$$y = r \times \sin(\theta_0 + i \times \theta_{step}) \tag{5}$$

$$x = r \times \cos(\theta_0 + i \times \theta_{step}) \tag{6}$$

where $\theta_0$ is the angle between the starting point and the *x*-axis; $\theta_{step}$ is the angle contained in each microline segment; and *i* ranges from 1 to $n_1$.

(2) Multiple-circular-arc discretization

The multiple-circular-arc discretization is mainly used for the drawing of multiple intersection basic-type curves in C-curves. The multiple-circular-arc discretization utilizes a small-line segment fitting interpolation algorithm and adds the interpolation control of the positive radius to the two semi-axes of the ellipse.

The multi-arc line fitting schematic is shown in Figure 2b; endpoints A, B, and tangent lines OA, OB in the plane is given. With A and B as endpoints, p3, p4, and p5 as tangents, four circular arcs are produced and fitted tangentially to OA and OB to obtain the AB multi-circular arc. Since OA is not equal to OB, and the arc AB approximates an elliptic arc, so the elliptic arc can be used to achieve the multi-circular-arc curve drawing in the algorithm.

In this algorithm, a $n_2$ ($n_2$ is an integer, $n_2$ and greater than 0) segment matrix is established, a positive circle radius $r_1$ and a non-uniform radius $r_2$ are set up, in order to fit the multi-circular-arc closer to the original garment contour curve, the $n_2$ segments are divided into upper and lower $n_2/2$ segments, the standard circle radius is set to 1, and the value of the new non-uniform radius for each segment is obtained by multiplying the coefficient of variation according to the garment contour curve modeling, as shown in Equation (9). After obtaining the radius of each segment, the sine theorem is used to calculate the angle of the center for each segment, the coordinates of the short and long half-axes of the ellipse can be obtained, and finally the coordinates of each *x*-axis and *y*-axis are transformed, shifted, and rotated to obtain the quarter-plus arc curve based on the center of the ellipse. As shown in Equations (7)–(10):

$$r_2 = r_1 \times \left[ 1 \pm \left( \frac{n_2}{2} - |i - 15| \right) \right] / \frac{n_2}{2} \times \varepsilon \tag{7}$$

$$\theta_i = \frac{i \times 3}{180} \times \pi \tag{8}$$

$$x_i = r_1 - r_2 \times \cos \theta_i \times \left( \frac{px}{r_1} \right) - (r_1 - px) + x_p \tag{9}$$

$$y_i = r_1 - r_2 \times \sin \theta_i \times \left( \frac{py}{r_1} \right) - (r_1 - py) + y_p \tag{10}$$

where, $r_1$ is the radius of the positive circle; $r_2$ is the non-uniform radius; $\varepsilon(\varepsilon > 0)$ is the coefficient of variation between the multi-circular arc curve model and the target curve; $\theta_i$ represents the angle of the center of each positive arc; $px$ and $py$ represent the lengths of the short and long semi-axes, respectively; $x_i$ and $y_i$ represent the coordinate values of the $n_2$ elliptical short and long semi-axes, respectively; $p$ represents the coordinate point of the center of the ellipse.

(3) Spline curve discretization

The spline curve discretization is mainly used for the drawing of S-curves, such as the sleeve head curve and side-slit curve. The data point and control point are two parameters that affect the fitting accuracy and fitting speed in the curve fitting process. Generally speaking, the fitting accuracy is proportional to the number of data points and control

points, and the fitting speed is inversely proportional to the number of data points and control points. Therefore, a good fitting method requires a large fitting accuracy with fewer t data and control points; the way to achieve this requirement is to insert different parameters and algorithms.

In summary, this study constructs a discrete function for spline curves based on coordinate arrays and coordinate matrices, and here the fitting algorithm for the sleeve head curve was used as an example.

According to the knowledge of the prototype, the key parameters were set up and the corresponding *x*-axis and *y*-axis coordinate formulas were calculated, and the top point of the sleeve head (shoulder point), the front underarm point and the back underarm point were set as the key parameter points of the curve according to the principle of drawing the sleeve head curve. By adding uniform data points to simulate a set of curve discrete point coordinates, the spline curve model was obtained. In order to improve the curve fitting accuracy, the curve was discretized into $n_3$ ($n_3$ is an integer, $n_3$ and greater than zero) control point coordinates, and the values of *x*-axis and *y*-axis of each coordinate point were traversed to generate the spline curve discretization model respectively.

The *x*-axis and *y*-axis arrays in the range of (0, $n_3$) were established, and the sleeve width was equally distributed into $n_3$-1 small segments, as shown in Equation (11).

$$x_i = i \times \frac{|P_6 P_7|}{n_3 - 1} \tag{11}$$

where, $x_i$ is the *x*-axis value of $n_3$ coordinate points; $|P_6 P_7|$ is the sleeve width.

The #Dataextraction command statement was used to obtain the coordinates of the intersection of each vertical line segment with the sleeve head curve and extract the *y*-axis values $y_i$ of each intersection and assign them to the *y*-axis array. The spline model was multiplied by the coefficient of variation to obtain the position of each coordinate point of the new curve. The coefficient of variation of sleeve width $v_x$ and the coefficient of variation of sleeve head height $v_y$ can be derived by parametric formulas. As shown in Equations (12)–(15):

$$v_x = \sqrt{(AH_{front} - y_{P_1})^2 + (AH_{back} - y_{P_1})^2} \tag{12}$$

$$v_y = \frac{B}{10} + 3 \tag{13}$$

$$\varepsilon_x = v_x / |P_2 P_3| \tag{14}$$

$$\varepsilon_y = v_y / S.T. \tag{15}$$

where, $AH_{front}$ is the front armhole length, $AH_{back}$ is the back armhole length; $B$ is the chest circumference; $v_x$ is the parameterization formula of sleeve width; $v_y$ is the parameterization formula of sleeve head height; $\varepsilon_x$ is the coefficient of change in sleeve width; $\varepsilon_y$ is the coefficient of change in sleeve head height.

The coordinate point matrix of the new curve can be constructed using the above formulas; $x_i$ and $y_i$ were multiplied by $\varepsilon_x$ and $\varepsilon_y$ respectively; and finally $n_3$ control point coordinates were returned, and the corresponding curve can be obtained after fitting the above coordinates.

## 3. System Design and Production

In order to verify the effectiveness of the garment curve fitting method based on the parametric design and to realize the automatic garment pattern generation system accordingly, this paper designed and built the system based on the Python language and the secondary development function of drawing software, taking a women's shirt in line with a certain enterprise style as an example.

### 3.1. Development Environment Construction

Jupyter Notebook was used to build the compilation environment. The friendly development environment allows the automatically generated parametric patterns to be compatible with external systems and facilitates parameter input, adjustment, and data storage. During the process of system development, third-party libraries such as numpy and math were imported, while the underlying logic required for pattern generation was implemented in conjunction with the contents of Section 2. The main purpose of secondary development is to take advantage of software combinations to obtain diversified functional development effects, simplify the drawing steps, and improve the stability of system operation.

Since the pattern-making software used by the company was AutoCAD, the secondary development of AutoCAD was demonstrated as an example in this study. It is worth pointing out that the system is not only applicable to AutoCAD; for example, Figure S1 in the Supplementary document shows a garment pattern prepared and based on other types of drafting software. This feature makes the system very adaptable to different companies with different pattern-making software, which greatly enhances the system's use scenarios.

### 3.2. Classification and Adjustment Methods of Parameters

In this study, the main types of parameters used in this paper include key parameters, secondary parameters, and variable parameters. The key parameters are the net dimensions measured by the human body. In the parametric design of women's shirts, the key parameters mainly include shoulder width, chest circumference, sleeve length, garment length, and collar circumference. Secondary parameters are obtained by the operation of key parameters according to the inherent structural relationship of the garment, such as back collar width, back collar depth, back width, etc. Variable parameters are modeling parameters in garment paper pattern design, such as the shape of waist curve, the shape of collar, the position and number of pockets, the position and number of buttons, etc.

Figure 3a shows the adjustment process of key parameters and related codes in the development environment. The adjustment of four key parameters, namely shoulder width, bust, sleeve length, and garment length, can acquire the basic shape of the shirt–body pattern (collar not included). Figure 3b shows the process of adjusting the sleeve width, a secondary parameter involved in the sleeve pattern and based on the key parameter of sleeve length. Figure 3c shows the process of adjusting the line of the waist area in the body pattern from a straight line to a curve, which also reflects the process of adjusting the variable parameters.

### 3.3. Establishment of Parametric Coordinate System

By using the pattern decomposition and fitting methods in Sections 2.1 and 2.2, a parametric coordinate system was established; Figure 4 shows the parameterized coordinate system of the sleeve pattern in the established coordinate system (the full data of the coordinate system is shown in the Supplementary document). Since the point coordinates, fitting algorithm, and body data in the coordinate system were all parametric variables, it is possible to obtain garment patterns of different body data and styles by modifying the variables in the system.

(a)

```
###############################
#Key Parameters
Pline_qianpian,Pline_houpian,Pline_fushi,spline_fushi_xy2,Pline_tiedai,lingwei,xiongwei,jingjiankuan,yichang,v_eg_long,v_qg_long,inm_long=yishen_jingyang(40,95,56,59)
point_a=APoint(0,740,0)
fushi_all=point_one_y_mirror(point_a,Pline_fushi,False)
```

(b)

```
def xiucha_zuo_xiuchang55(xiuchang):
    spline_xiucha_xy =np.zeros(7*3,dtype=np.double)
    spline_xiucha_xy=np.array([300,-600,0,300,-475,0,
                               312.5,-465,0,325,-475,0,
                               325,-500,0,350,-500,0,
                               350,-600,0
                               ])
    spline_xiucha_xy = aDouble(spline_xiucha_xy)
    Pline_xiucha = pyacad.model.AddPolyLine(spline_xiucha_xy)
    point1=APoint(300,-600,0)
    point2=APoint(350,-600,0)
    Pline_xiucha.Closed = True  # 设置多段线闭合
    return Pline_xiucha
```

(c)

```
#Arc of the waist
fanling_all=point_one_y_mirror(point_a,Pline_fanling,False)
#
Pline_fanling_all= pyacad.model.AddPolyLine(fanling_all)
Pline_fanling_all.Closed = True
fanling_all_offset_np=polyline_offset(Pline_fanling_all,distance=-10,biaozhi=False)
Pline_fanling_all.Delete()
Pline_fanling.Delete()
```

**Figure 3.** Schematic diagram of each type of parameter adjustment process. (**a**) key parameters; (**b**) secondary parameters; (**c**) variable parameter.

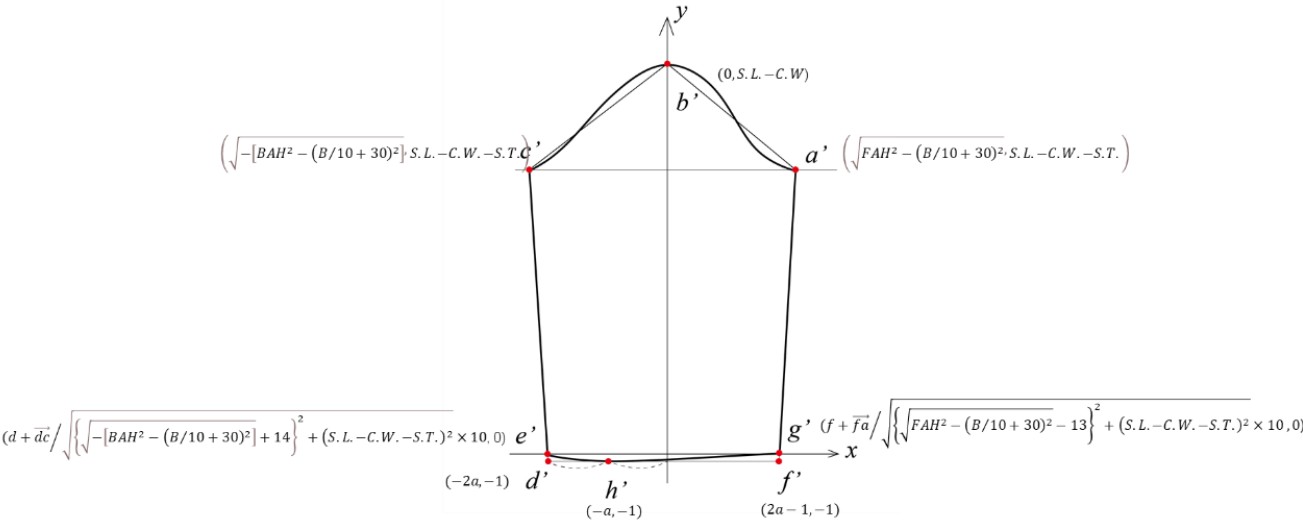

Note: where a is the variable, N=neck circumference, L=length of clothing, b=bust circumference, FAH=front armhole length, BAH=back armhole length, S.L.=sleeve length, C.W.=sleeve head width, S.T.=sleeve hill height

**Figure 4.** The parameterized coordinate system of the sleeve pattern.

In the automatic generation process of the sleeve pattern, the sleeve length ($S.L.$) is the key parameter mentioned in Section 3.2; points a', b', c', e', and g' in the figure are secondary parameters, and the position parameters of these coordinate points all contain $S.L.$; points d', h', and f' are variable parameters that control the width of the cuff, and their positions are mainly determined by the variable $a$. The shape of the cuff can be controlled by controlling the size of $a$. As shown in Figure 5.

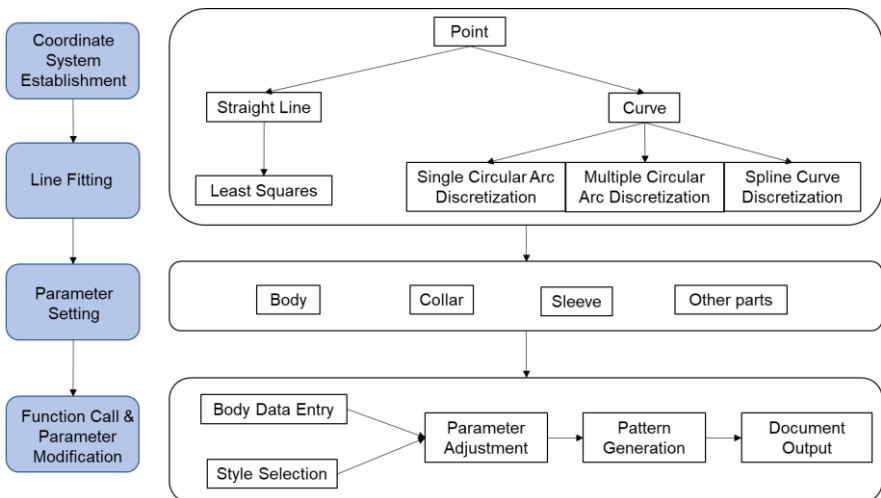

**Figure 5.** Parametric model framework of the system.

### 3.4. Parametric Algorithm Model Building

A Python3 development environment was established in this study based on the Windows system and AutoCAD2018 software, and the algorithm was compiled in Jupyter Notebook. The plug-in ArxDbg was automatically loaded before the algorithm was compiled to realize the control and testing of AutoCAD software. The parametric model framework was mainly divided into three major layers, as shown in Figure 4.

The above model sets the five key parameters of chest circumference, shoulder width, garment length, sleeve length, and collar circumference as interactive parameters that can be controlled by the user, and the program will automatically calculate the coordinates of each control point according to the parametric algorithm model to generate a new pattern according to the user-set values during operation. According to the customized requirements, the interactive parameters can be extended from key parameters to secondary parameters or variable parameters to further improve the accuracy of parametric plate making.

For example, by adding an interactive parameter of "width of depression towards the middle (Wm)" to the above algorithm model, the user can control the tightness of the shirt waist by customizing the size of Wm. Figure 6 shows the comparison effect of the back piece with different Wm values.

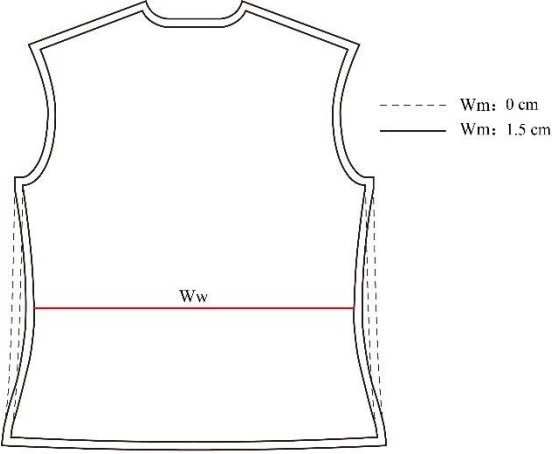

Wm: Width of depression towards the middle
Ww: Waist width after adjustment

**Figure 6.** Parameter constraint relationship between waist width and waist circumference.

In order to ensure the rationality and accuracy of the pattern, it is necessary to establish a constraint relationship between parameters with correlation; for example, in the process of the personalized adjustment of the line of the waist area, the width of the adjusted waist width should not be less than one-half of the waist circumference, otherwise the finished garment cannot be worn properly, as shown in Figure 6.

### 3.5. Lightweight Main Frame System and Code Encapsulation

In order to meet the readability requirements, the naming of all pattern functions and parameter expressions was written in accordance with the garment pattern mapping requirements. The code encapsulation can use the parameters as variables inside the function to process the data between the parameters and can also call any other functions and pack them into individual code blocks to complete the function building according to the operation logic. After all the functions are built, different classes can be defined (classes) based on style types and curve types, and the inheritance mechanism can be used to reuse the code.

At the same time, by using the Python language, the features of external files and variables can be imported, and different types of parametric garment pattern models can be compiled into py. files and encapsulated and called in the system, which can realize the lightweight processing of the main framework, reduce memory loss, and speed up pattern generation.

### 3.6. Design and Implementation of Front-End Interface

The front-end interface was designed and developed using the Vue framework and presented as a web or applet, as shown in Figure 7. In the front-end interface, dimensions such as collar circumference, bust circumference, waist circumference, garment length, sleeve length, and sleeve head width can be entered, and a certain style of garment can be selected. The automatically generated patterns were saved in the DXF file format, which was available for downloading or sent to the production side.

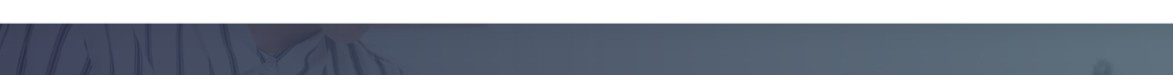

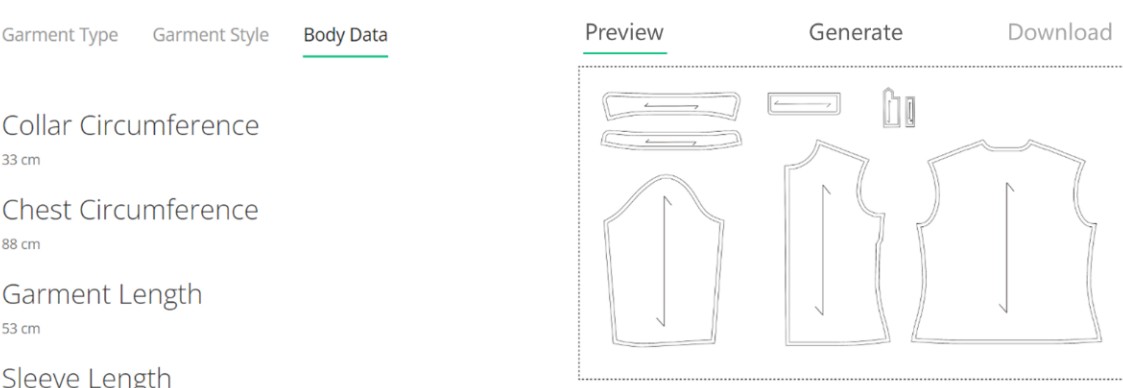

**Figure 7.** Front-end interface of the system.

## 4. Experiments and Results

In order to verify the validity and accuracy of the garment pattern generated by the system, the system validation experiments are conducted from both pattern comparison experiments and virtual tests.

### 4.1. Pattern Comparison Test

The pattern comparison test was conducted between the pattern made by the system proposed in this study and the pattern made manually by the employees of the company with rich experience in pattern making. The number of the automatically acquired pattern was group A, and the number of the manually made pattern was group B. The validity and accuracy of the patterns generated by the system were evaluated by comparing the similarity of the two groups of patterns under the same parameters.

#### 4.1.1. Participate

A total of 53 healthy female participated in the experiment. Their (M ± SD, mean and standard deviation) age, height, and weight were 27.6 ± 6.8 years, 165.0 ± 10.4 cm and 55.2 ± 11.0 kg, respectively. Before the experiment, all the participants were informed about the details of the experiment beforehand and their body data were measured using three-dimensional body scanning equipment (BOK, Body 3D Scanner II, China) with an accuracy of ±0.5 mm. Information on parameters such as chest circumference, collar circumference, waist circumference, and height were extracted from the participant's three-dimensional model, as detailed in Table S1 in the Supplementary document. Figure 8 shows the processing of a three-dimensional mannequin of a particular participant.

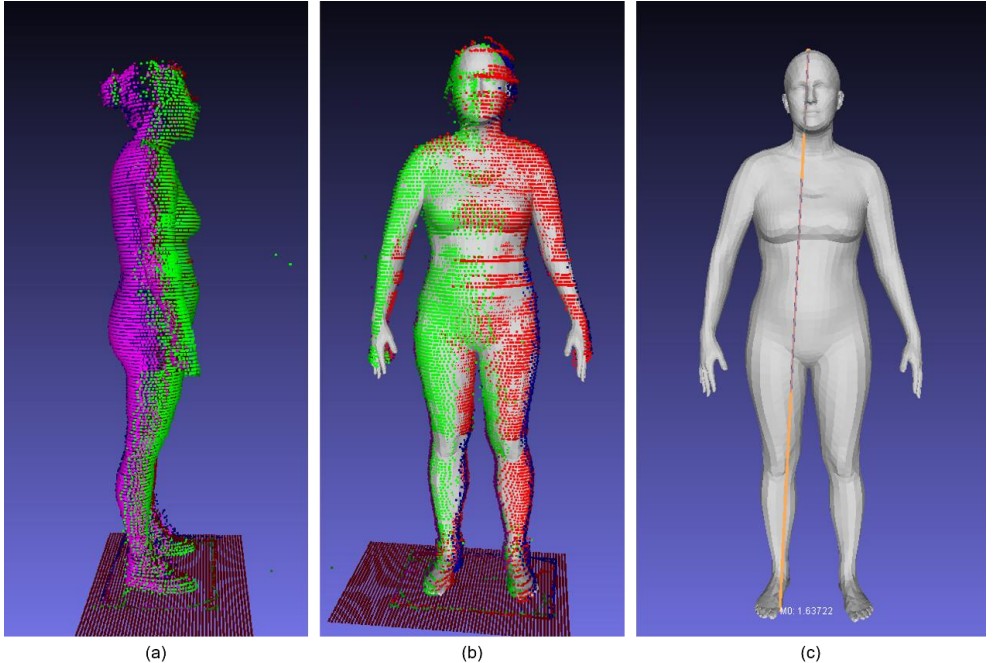

**Figure 8.** (**a**) Scanned human body point cloud data; (**b**) human body point cloud model after data pre-processing; (**c**) model for exporting human body data.

#### 4.1.2. Experimental Procedure

Group A: A total of 53 participants' body parameter data were input in the above system, and two different styles of women's shirt patterns were generated according to each group of data. These two different patterns of women's shirts are shown in Figure 9. Their specific specifications are detailed in the Supplementary document.

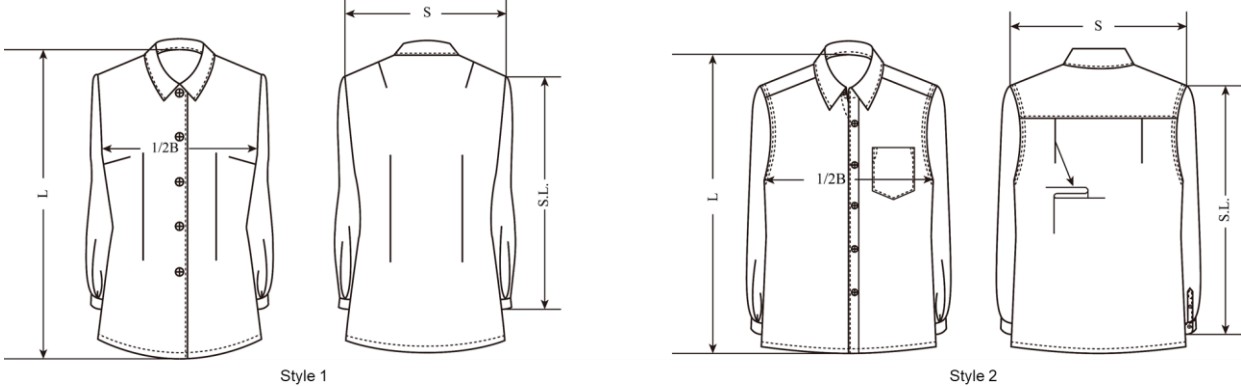

Style 1

Style 2

Note: L=length of clothing, B=bust circumference, S=shoulder width, S.L.=sleeve length

**Figure 9.** Comparison of two different styles of women's shirts.

Group B: Since the preset garment style of group A was consistent with the garment style in the customization business of a certain enterprise, the employees of the enterprise were invited to make the patterns manually using the AutoCAD software. The parameters and quantity of the patterns were the same as those of group A.

Since the parameters of the corresponding patterns in the two groups were the same, i.e., the point coordinates of the patterns and the straight-line fitting method were the same, the comparison of the similarity between the two groups of patterns was carried out mainly from the perspective of curve fitting. Taking the side curve, front 1/4 of ellipse, back 1/4 of ellipse, sleeve head curve, and armhole curve as examples, the five curves in each pattern were extracted separately and exported as jpg files. The curves with the same parameters and the same type were grouped. Figure 10 shows the curves extracted from the pattern. Before comparing, the same group of curves were pre-adjusted to keep the consistency of size, color, and other parameters between the two groups of images.

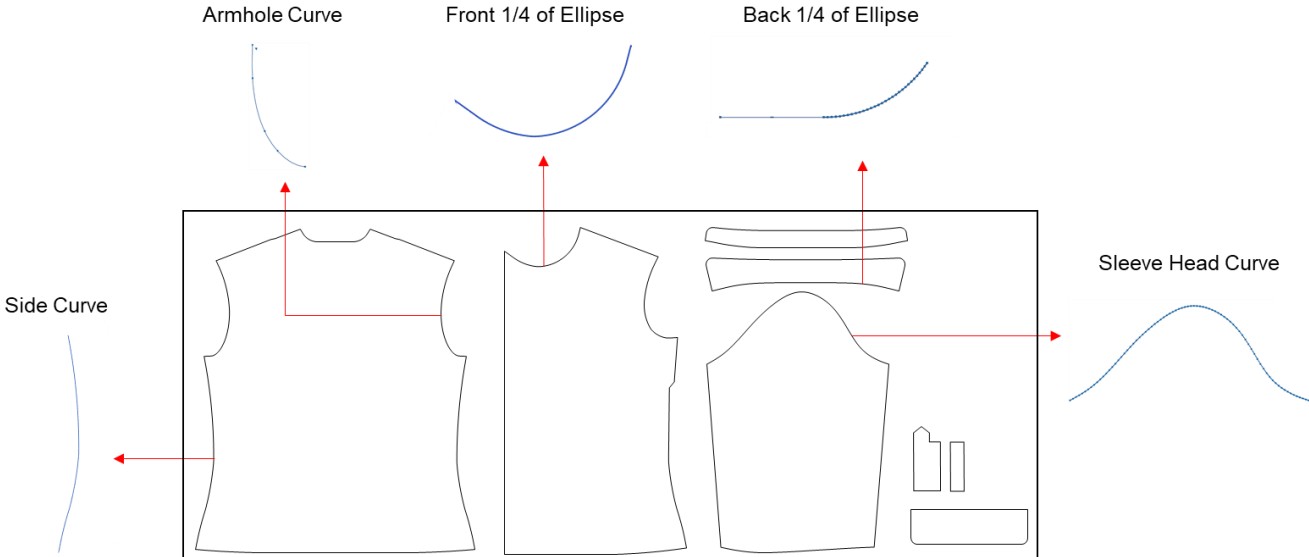

**Figure 10.** Schematic diagram of the curve extracted from the pattern.

A similarity index $R_{NL}$ was defined, and the similarity test for curve fitting can be performed by using the similarity-index-of-label-graph library in Python. Under the condition of the same parameters, the fitted value $A_i$ of group A pictures and the fitted value $B_i$ of group B pictures of the same category were considered as points in N-dimensional space, and the closer $A_i$ and $B_i$ were (the closer the value of $R_{NL}$ was to one), the higher

the similarity of the two patterns. As shown in Equation (16). Figure 11 shows a schematic diagram comparing the fitted similarity of the sleeve head curves for a group of patterns.

$$R_{NL} = 1 - \sqrt{\frac{\sum (A_i - B_i)^2}{\sum A_i^2}} \tag{16}$$

where: $R_{NL}$ is the similarity index; $A_i$ is the fitted value of the $i$th image in group A; $B_i$ is the fitted value of the $i$th image in group B.

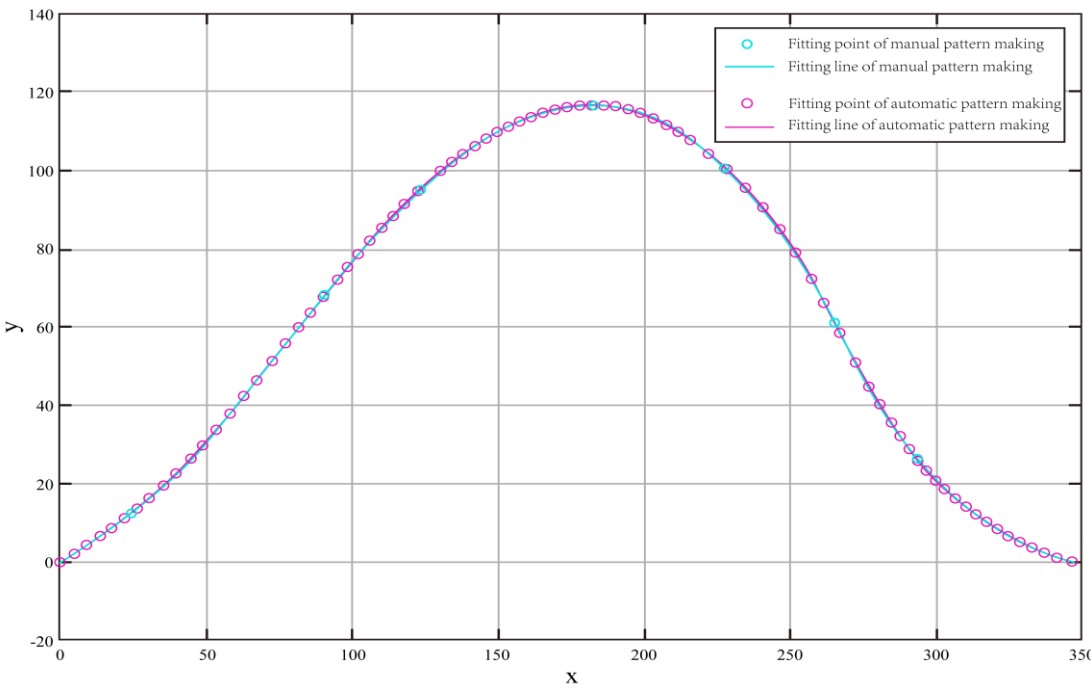

**Figure 11.** Comparison of similarity of fit of sleeve head curves.

### 4.1.3. Experimental Results

After the comparison, the average of the test values for the comparison of similarity between the pattern pictures of group A and group B is shown in Figure 12. The worst mean similarity among all the data was the sleeve head curve of style one with a test value of 0.9873, because the complexity of the sleeve head curve is higher than the other curves, and more parameters need to be substituted in the fit.

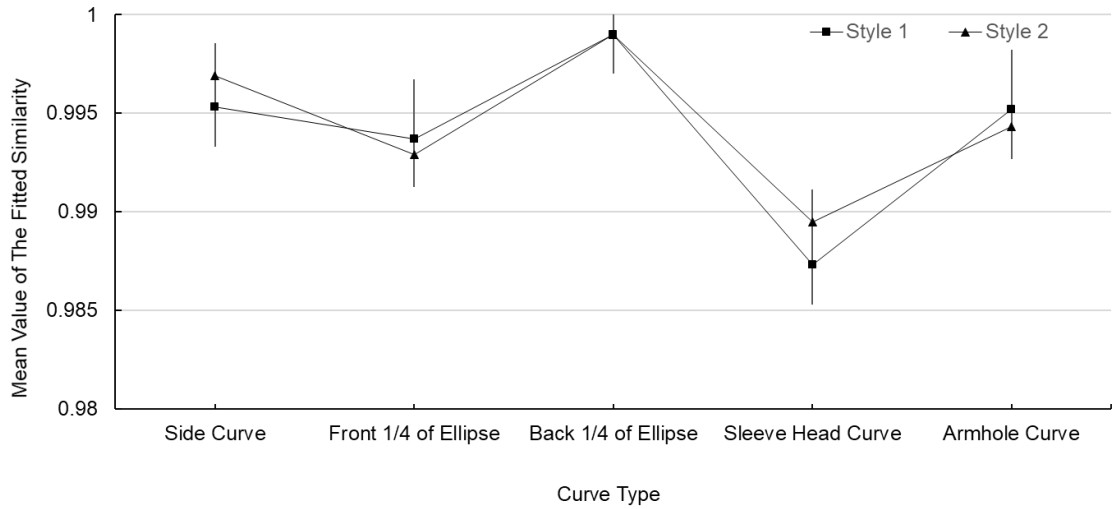

**Figure 12.** The mean value of the fitted similarity of each curve.

*4.2. Virtual Testing*

With the rapid development of three-dimensional garment modeling technology, technologies such as virtual modeling and virtual fitting provide new methods for objective assessments of garment wearability. This study proposes to evaluate the comfort and fit of garment patterns after performing virtual sewing using virtual pressure testing and testing of the air layer under the garment.

4.2.1. Construction of the Solid Model

In this study, the scanned human body model (with a sampling accuracy of 0.5 mm) in Section 4.1.1 was constructed as a solid model (average modeling accuracy of 0.5 mm) based on Geomagic Studio software and reverse engineering methods. Geomagic Studio can easily create polygonal mesh models based on the point cloud data from three-dimensional human body scans and quickly and automatically construct human body NURBS (Non-Uniform Rational B-Splines) surfaces, which are suitable for three-dimensional human body modeling. The scanned human body point cloud file was imported into Geomagic Studio software to remove the noisy points outside the body to improve the accuracy of the subsequent model. As shown in Figure 13.

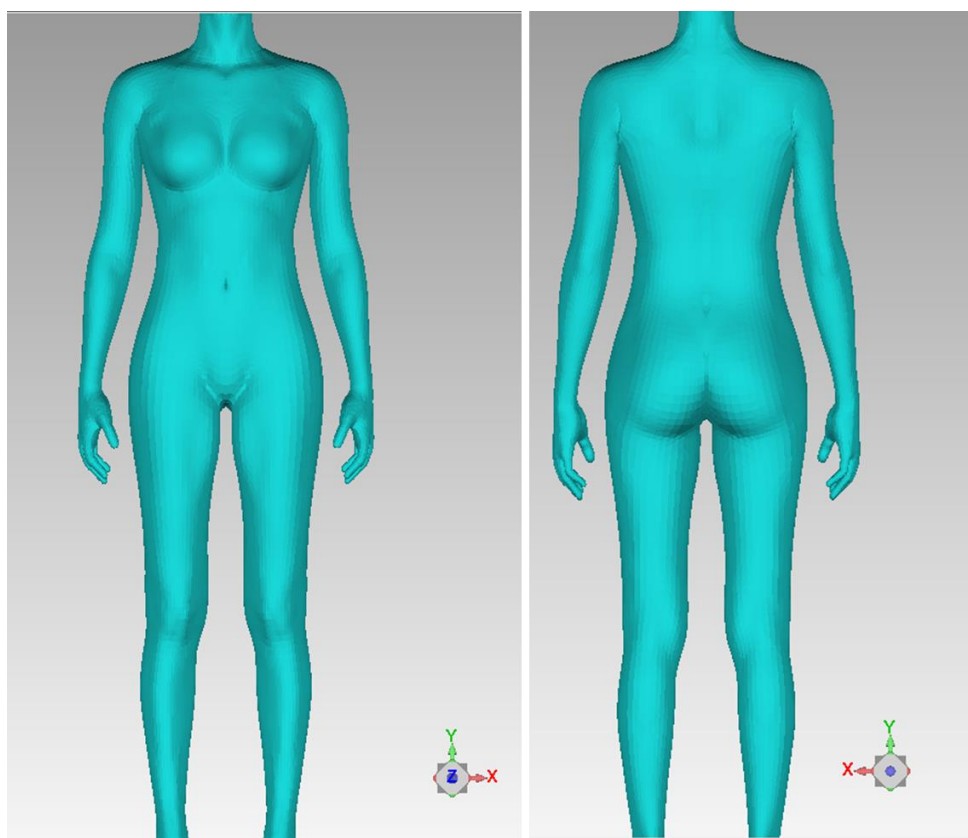

**Figure 13.** Solid model of human body after construction.

4.2.2. Virtual Pressure Test

The created solid model of the human body and the automatically generated patterns were imported into the CLO 3D software, and three different groups of fabric properties were set in the software, i.e., 100% cotton, 100% polyester, and 50% cotton + 50% polyester, with the specific index parameters shown in Table 2. The test diagram is shown in Figure 14a.

**Table 2.** Relevant parameters of the three fabrics.

| No. | Indicator | Grammage/(g/m²) | Thickness /(mm) | Internal Damping | Friction Coefficient | Elongation/(%) | | Bending Stiffness/ (gf·cm²/cm) | |
|---|---|---|---|---|---|---|---|---|---|
| | | | | | | Warp | Weft | Warp | Weft |
| 1 | 100% cotton | 146.7 | 1.25 | 0.0001 | 0.03 | 7.5 | 7.5 | 0.030 | 0.030 |
| 2 | 100% polyester | 119.0 | 1.10 | 0.0001 | 0.03 | 6.1 | 6.1 | 0.016 | 0.016 |
| 3 | 50% cotton + 50% polyester | 78 | 1.17 | 0.0001 | 0.03 | 3.4 | 3.4 | 0.074 | 0.074 |

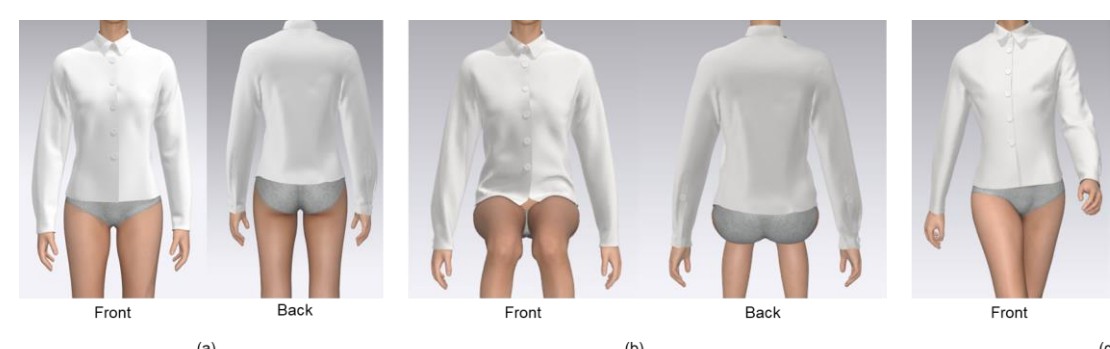

**Figure 14.** Display page of virtual model. (**a**) Standing posture; (**b**) sitting posture; (**c**) walking posture.

The garment pressure test is an important indicator used to evaluate the comfort of garments. In this paper, the pressure test tool in the CLO 3D menu was used to measure the pressure values of the garment in the standing, sitting, and walking states of the virtual model to verify the comfort of the pattern, as shown in Figure 14.

At the same time, the pressure parameter values of the collar, bust, waist area, and sleeve hole areas of the virtual model were extracted in CLO 3D according to the female body structure and the key areas that affect the fit of the garment, as shown in Figure 15.

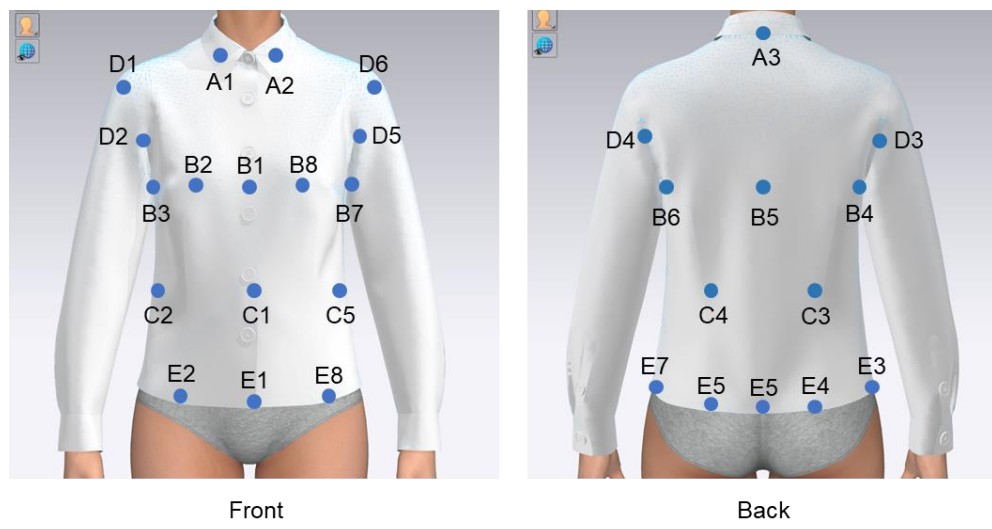

**Figure 15.** Position of the pickup point for the stress test.

### 4.2.3. Air Layer under the Garment Testing

In order to better characterize the fit between the sewn garment and the body, the determination of the air layer under the garment was carried out. The sewn garment model and the solid model of the human body were imported into the Geomagic Wrap software to test the distribution of the air layer between the garment and the human body, where green indicates a small air layer thickness and red indicates a large air layer thickness.

### 4.2.4. Test Results

(1) Virtual pressure

The pressure distribution map can reflect the compression of the garment on human soft tissues by color according to the degree of force. The red area indicates that the pressure of the garment on the soft tissues of the body is greater; the blue area is the opposite. It is generally believed that the maximum pressure comfort threshold for women's upper body is 2.753kPa [28], and when the pressure value is less than this value, the comfort of the garment is better.

Figure 16 shows the schematic diagram of the virtual pressure test. The pressure map was colorless overall and blue locally, which indicates that the overall pressure of the garment was low. Meanwhile, the results of the pressure values of the neckline, chest, waist, sleeve holes, and hem of the virtual model are shown in Table 3. From the direction of pressure measurement parts, the pressure values of the chest, waist, and hem were smaller, and the pressure values of the collar and sleeve holes were larger; from the direction of fabric properties, the pressure values of fabrics with higher grammage and thickness were larger for the human body; from the direction of body posture, the pressure values of garments on the human body were smaller in standing and sitting postures, and the pressure values of garments in walking postures were larger. The measured pressure values were kept within 2.753 kPa, and the experimental results showed that the garment patterns generated by the system had good comfort.

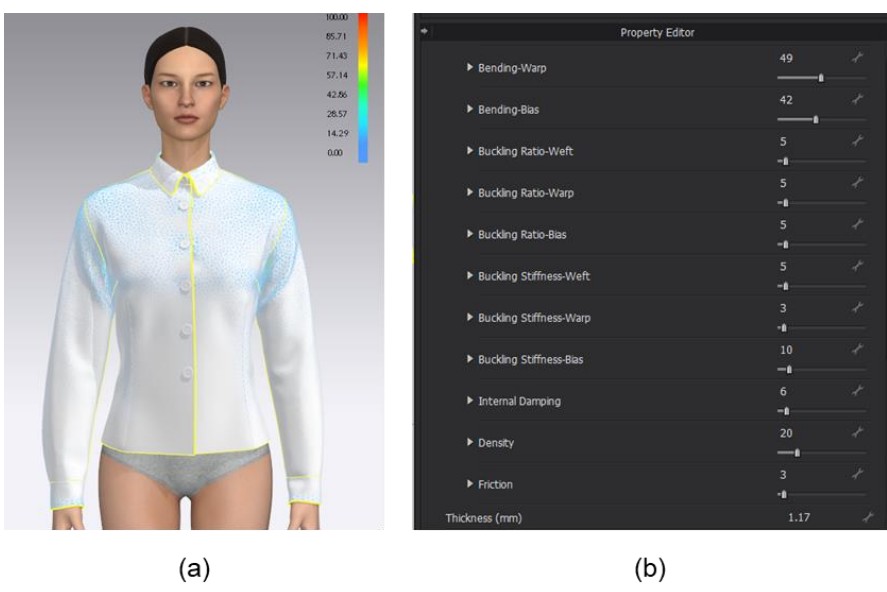

(a)  (b)

**Figure 16.** (**a**) Pressure test chart; (**b**) mechanical properties of the fabric.

(2) Thickness of air layer under the garment

The thickness of the air layer between the garment and the body can characterize the degree of fit between the garment and the body. Green indicates that the thickness of the air layer in the region is small, and red indicates that the thickness of the air layer in the region is large, as shown in Figure 17. Table 4 shows the maximum value, average values, and standard deviation of the air layer thickness under the garment for each part extracted from all models. In the shoulder and chests, clothing and the body are maintained at a better fit, while in the body curve rate of the larger parts, such as the waist, the thickness of the air layer under the clothing is larger, which is also consistent with the fit of clothing and the body in daily life.

**Table 3.** Average of the pressure values of each part.

| Category | Fabric | Collar/(kPa) | Bust/(kPa) | Waist/(kPa) | Armhole/(kPa) | Hem/(kPa) |
|---|---|---|---|---|---|---|
| | 1 | 0.45~1.13 | 0.31~0.89 | 0.13~0.65 | 0.55~1.13 | 0.21~0.71 |
| Standing posture | 2 | 0.38~1.27 | 0.41~0.63 | 0.22~0.43 | 0.44~1.09 | 0.29~0.51 |
| | 3 | 0.36~1.21 | 0.28~0.59 | 0.19~0.45 | 0.40~1.03 | 0.20~0.49 |
| | 1 | 0.49~1.06 | 0.53~0.72 | 0.15~0.47 | 0.25~0.99 | 0.23~0.53 |
| Sitting posture | 2 | 0.35~1.25 | 0.33~0.64 | 0.21~0.53 | 0.24~0.99 | 0.29~0.49 |
| | 3 | 0.33~1.31 | 0.28~0.61 | 0.19~0.45 | 0.22~0.91 | 0.23~0.43 |
| | 1 | 0.83~1.95 | 0.67~1.21 | 0.46~0.97 | 1.05~1.49 | 0.53~0.83 |
| Walking posture | 2 | 0.55~1.83 | 0.63~1.19 | 0.44~0.93 | 1.01~1.44 | 0.52~0.79 |
| | 3 | 0.67~1.81 | 0.58~1.13 | 0.45~0.88 | 1.02~1.41 | 0.51~0.73 |

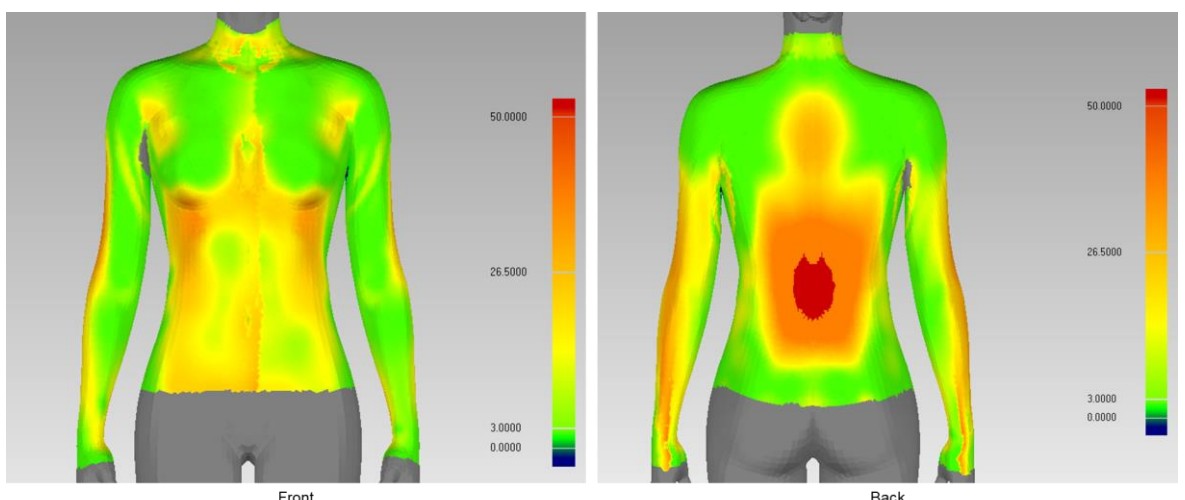

**Figure 17.** Testing of the thickness of the air layer under the garment.

**Table 4.** Thickness of the underclothing air layer in each part.

| | Neck/(mm) | Shoulder/(mm) | Chest/(mm) | Upper Arms/(mm) | Lower Arms/(mm) | Waist/ (mm) | Total/ (mm) |
|---|---|---|---|---|---|---|---|
| Maximum value | 19.2865 | 8.7765 | 11.2334 | 24.6589 | 28.3312 | 54.7128 | 54.7128 |
| Average value | 11.7765 | 4.2311 | 5.8897 | 17.5567 | 15.8774 | 25.8879 | 12.6992 |
| Standard deviation | 2.9865 | 1.7743 | 2.1159 | 7.4469 | 9.8863 | 15.6985 | 9.3506 |

## 5. Conclusions

Based on the design concept of the parametric design, this paper proposed a method to automatically generate garment patterns. By decomposing the garment pattern, the decomposed pattern was represented on a two-dimensional plane with parametric variables. According to the classification of common curves in garment patterns, three curve-fitting algorithms based on different parameters were derived and combined with the Python language to propose the underlying logic of automatic pattern decomposition and fitting, and further implement an automatic garment-pattern-generation system that can adapt to different drawing software. By classifying the parameters needed for drawing garment patterns, the function of using a small number of key parameters to constrain secondary parameters was realized, and the input of variable parameters with highly interactive performance was realized in combination with a front-end page. Meanwhile, in cooperation with a company, an automatic pattern-generation system for women's shirts was developed

to match the design style of the company. Three different methods are proposed to verify the accuracy of the garment pattern from the perspective of curve fitting similarity and three-dimensional virtual modeling.

There are still shortcomings in this study. One of the shortcomings is that the participants of the test are all young women, and no test was conducted for the elderly; another is that the import of mannequins in the system is still manual, therefore further research should be devoted to solving the problem of automatically reading the data of scanned mannequins and importing the data into the pattern making system to achieve higher integration.

**Supplementary Materials:** The following supporting information can be downloaded at: https://www.mdpi.com/article/10.3390/su15021268/s1, Figure S1: Patterns drawn after secondary development of other mapping software; Figure S2. Parameterized coordinate points of the original model; Figure S3. Structure diagram of style 1 shirt; Figure S4. Structure diagram of style 2 shirt; Figure S5. Style 1 pattern diagram for participants No. 1–4; Table S1: Parameterized coordinate points of the original model; Table S2. Baseline body size data of the participants; Video S1: Initial prototype pattern; Video S2. Adjustment of key parameters; Video S3. Linkage control between key parameters and other parameters.

**Author Contributions:** Conceptualization, R.Z.; Methodology, P.J. and J.F.; Software, P.J.; Validation, L.L. and H.Z.; Formal analysis, Q.C. and L.L.; Investigation, R.Z.; Resources, Q.C.; Data curation, P.J. and R.J.; Writing—original draft, P.J.; Writing—review & editing, J.F. and R.Z. All authors have read and agreed to the published version of the manuscript.

**Funding:** This work was supported by Natural Science Foundation of Shanghai: [Grant Number 21ZR1400100]; Arts and Humanities Research Council: [Grant Number AH/T011483/1]; Shanghai Style Fashion Design & Value Creation Collaborative Innovation Center: [Grant Number ZX201311000031].

**Institutional Review Board Statement:** Not applicable.

**Informed Consent Statement:** Informed consent was obtained from all subjects involved in the study.

**Data Availability Statement:** Not applicable.

**Conflicts of Interest:** The authors declared no potential conflict of interest with respect to the research, authorship, and/or publication of this article.

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
