# Peer review of "Design and Research of Automatic Garment-Pattern-Generation System Based on Parameterized Design"

_sustainability, doi:10.3390/su15021268_

Round 1

Reviewer 1 Report

General

In this manuscript, an automatic garment pattern generation method was proposed, based on the design concept of parametric design. Three curve fitting algorithms based on different parameters were deduced according to the classification of common curves in the pattern, and combined with Python language and secondary development function of AutoCAD to realize the automatic generation of different patterns.

The proposed research can be caracterised as a case study, considering that the developed algorithms are applicable only for the construction of the basic pattern of a women's shirts, and that the possibility of pattern drawing is related to the capabilities of the AutoCAD software. The similar investigation was published in 2001.: Petrak, S. & Rogale, D.: Methods of automatic computerized cutting pattern construction, available at https://www.emerald.com/insight/content/doi/10.1108/09556220110396542/full/html.

It would be correct if the mentioned work was cited, considering the very similar methodology.

Abstract

„The results show that the pattern generated by the system was extremely similar to the enterprise's manual pattern, and the system parame-ters can be adjusted according to the body shape data, the type or style of garment to obtain the garment pattern required by the enterprise.“

R:  Nowhere in the manuscript isn't presented how the system parameters can be adjusted according to data on body shape, type or style of clothing. Need to explain in more detail and present the results for this part.

„Since the point coordinates, fitting algorithm and body data in the coordinate system were all parametric variables, it is possible to obtain garment patterns of different body data and styles by modifying the variables in the system.

R:  It is necessary to explain in more detail how and which styles modifications the system enables.

Section 3.3.

Figure 4. Parametric model framework of the system.

R:  „Body Data Entry“ - It is not explained what body data was entered. It need to be explained more in details.

„Style Selection“ - It is not explained what types of styles can be selected? How different they can be compared to the basic pattern?

Section 4.1.

„The pattern comparison test was conducted between the pattern made by the system proposed in this study and the pattern made manually by the employees of the company with rich experience in pattern making.  The validity and accuracy of the patterns generated by the system were evaluated by comparing the similarity of the two groups of patterns under the same parameters.“

R: The pattern made manually with which the comparison was made are not shown in the paper. It is not stated what was specifically compared on the patterns, which variables?

Section 4.1.1.

„A total of 20 healthy female participated in the experiment.“

R: The sample is too small to carry out relevant statistical analysis. Also, did you analyzed participants according to body type?

Section 4.1.2.

„…two different styles of women's shirt patterns were generated according to each group of data, totaling 40.“

R:  What is the difference between two styles of women's shirt patterns? Technical drawings of the model and accompanying descriptions are not presented.  How much more complex is the second model than the basic pattern?

„Figure 7. Schematic diagram of the curve extracted from the pattern.“

R:  The figure has poor resolution and is not clearly explained.

Section 4.2.1.

 „The virtual model body type data of the above 20 subjects were imported in the editor, respectively.“

R:  How accurately is the parametric body model adapted to the different body types in the sample, given its limitations in the number of measures that can be adapted, as well as the shape of body parts?

„Figure 9. (a) Display page of virtual model; (b) Pressure test chart of the same model with differ-ent fabrics.“

R:  Did you determine the same results on all test subjects in the sample by analyzing the pressure?

The results of fit evaluation shouls be disscussed regarding different body types and posture and also for different dynamic positions for comparison and better explanation of simulation results, since the verification with the real prototype is missing.

Section 4.2.2.

„The red area indicates that the pressure of the garment on the soft tissues of the body is greater; the blue area is the opposite.“

R:  The pressure distribution should be explained regarding the applied values of fabric mechanical properties.

„It is generally believed that the maximum pressure comfort threshold for women's upper body is 2.743kPa, and when the pressure value is less than this value, the comfort of the garment is better.“

R:  Please provide a source for this claim. Does this claim have a basis in a scientific researches, especially considering the influence of a complex mechanical and physical fabric properties on model fit?

Given that the presented research is quite complex with a numeros experts included in development and evaluation process, there is a question how complex is the system adaptation process  for another, more copmlex garment? And also for user’s with a different body types, espesialy older population?

There are also some technical errors in the manuscript.

Reviewer 2 Report

This manuscript proposed a methodology for generating garment pattern from curve fitting algorithms. Basically, this manuscript is well written and organized. It is suggested (1) to provide a more comprehensive description of the state-of-the-art of garment pattern generating techniques, and (2) to enhance the conclusion section so that it reflects the four challenges described at the end of Section 1.

Reviewer 3 Report

1.Using CLO to measure the virtual pressure can only get the pressure change trend, and can not reflect the fit.How do you get the pressure at kept within 2.743kPa is fit or comfortable?

2. There are many articles on parametric design of clothing patterns, which are not cited. How do you judge the differences and similarities between your research and those of others? How do you know the progressiveness of your research?

4. Many links in the article are not clear.

Reviewer 4 Report

The topic is actual and interesting. The authors present an alternative solution to obtain the patterns of a woman blouse by parametric design. Interesting research papers have already been published in the field of parametric design for the fashion and apparel industry by Pascal Bruniaux, Sybille Krzywinski, Zoran Stjepanovic, etc.

The authors of this article present a different approach to the parametric design of a woman blouse by using AutoCAD, a software generally used in mechanical engineering.

I kindly recommend to clarify the following aspects:

1. specific technical terms: "sleeve cap curve" replace with" sleeve head line/curve"; "sleeve fat width" replace with "sleeve width"; "sleeve hill height" replace with "sleeve head height"; "sleeve hill" replace with "sleeve head", etc. I have noticed some inconsistency in the naming of the various lines of the pattern contours, which is not correct. I can recommend you to read some books by Winifred Aldrich. I recommend that you check all the names of the contour lines of the garment.

2. it is said " design concept of parametric design, a method of automatic garment pattern generation based on parametric variables and Py-thon language was proposed in this paper, in order to solve the problem of wasted resources and costs caused by manual pattern making in garment personalization. "...

I suggest to review this statement from the following  reasons: 

-the patterns in the clothing companies are designed with special software (Lectra, Optitex, Gemini, Gerber, Graphis, etc.). The CAD's for the fashion and clothing industry have special tools and instruments that allow the model to be personalised and individualised according to the client's wishes. When it comes to haute couture, i.e. very complex models made of different materials, the patterns are made in a special way (moulage and draping method);

-it is known that there are special tools (CAD's software) which allow the garment design personalization (Lectra, Gemini, Graphis, etc.). With the tools of CAD a designer has the possibility to explore different design solutions for the selected model, to design many models with accuracy and in a reasonable time (the design phase is a sustainable phase). 

-it is known that the shape of a garment can be drawn in the software Inventor. The user has the option of editing an Excel file linked to the Inventor file, in which he or she specifies the input data and all the necessary mathematical relationships to determine the shape and size of the garment.

3. I am trying to find some information about the solution used to create the patterns of the garment pieces. It is important to determine the position of the main points of the pattern outline and then these points are connected with straight or curved lines.

4. table 1, page 4.... the back neckline is known as a 1/4 of ellipse( in all technical materials on garment pattern design);

5. explain the following abbreviations: ET, PDS, CNC, VUE, RNL;

6. in section 4.1.1 it is said " 66.7±5.9 years, 166.3±7.5 cm and 55.6±10.3 kg, respectively. " Why are these data important for the proposed parametric design solution?  I have seen the values of some tollerances: How were these calculated? These values have nothing to do with interdimensional anthropometric intervals.

7. it is not feasible to analyse and compare the patterns created manually and those created with IT. I recommend that you compare the patterns created with parametric design solutions with the patterns created with the tools of CAD. It is obvious that any pattern created with the specific tools of IT can be designed in less time than a manual pattern.

8. page 13....It is obvious that any pattern created with the specific tools of IT can be designed in less time than a manual pattern. I recommend to revise the content of the table 2. 

9. in section Conclusions it is said " The experimental results show that the patterns automatically generated by the system were highly consistent with those manually produced by the enterprise, and could significantly reduce the labor cost and time cost of the enterprise.  By modifying the parametric variables in the system, a new model can be generated and called on the basis of the original parametric model to meet the needs of garment enterprises for the automated generation of personalized pat-terns for different body types and different types of garments." I recommend that you revise these statements in relation to all my previous comments. 

Round 2

Reviewer 4 Report

The authors have taken my recommendations into account.

It is an improved version.